# Examination of Upper Limb Function and the Relationship with Gross Motor Functional and Structural Parameters in Patients with Spinal Muscular Atrophy

**DOI:** 10.3390/biomedicines11041005

**Published:** 2023-03-24

**Authors:** Aleksandra Bieniaszewska, Magdalena Sobieska, Barbara Steinborn, Ewa Gajewska

**Affiliations:** 1Department of Developmental Neurology, Poznan University of Medical Sciences, 60-355 Poznan, Poland; bstein@ump.edu.pl (B.S.); ewagajewska1011@gmail.com (E.G.); 2Doctoral School, Poznan University of Medical Sciences, Bukowska 70, 60-812 Poznan, Poland; 3Department of Rehabilitation and Physiotherapy, Poznan University of Medical Sciences, 61-545 Poznan, Poland; msobieska@ump.edu.pl

**Keywords:** spinal muscular atrophy, upper limb assessment, functional assessment, structural parameters, physiotherapy assessment

## Abstract

Spinal muscular atrophy (SMA) is an autosomal recessive disorder caused by the deletion or/and mutation in the survival motor neuron 1 (SMN1) gene on chromosome 5. Until now, only a few articles investigating the relationship between upper limb function and the gross motor function in untreated SMA patients have been published. However, there is still a lack of publications including the relationship between structural changes such as cervical rotation, trunk rotation and side trunk shortening, and upper limb function. The aim of the study was to examine the upper limb function in patients with spinal muscular atrophy and the relationship between the upper limb function, gross motor function, and structural parameters. We present an analysis of 25 SMA patients, divided into sitter and walker groups, undergoing pharmacological treatment (nusinersen or risdiplam), examined twice between the initial examination and evaluation after a 12-month period. The participants were tested using validated scales such as the Revised Upper Limb Module (RULM), the Hammersmith Functional Motor Scale—Extended (HFMSE), and the structural parameters. Our results showed that patients demonstrated greater improvement on the RULM scale than on the HFMSE scale. Moreover, persistent structural changes negatively affected both the upper limb function and gross motor skills.

## 1. Introduction

Spinal muscular atrophy (SMA) is an autosomal recessive disorder caused by deletion or/and mutation in the survival motor neuron 1 (SMN1) gene on chromosome 5 [1,2]. Mutation in the SMN1 gene results in insufficient production of the survival motor neuron (SMN) protein and degeneration of motor neurons in the spinal cord and the brain stem [3]. This leads to muscle weakness, contractures, deformities of the spine and chest, and respiratory disorders, which significantly reduce the quality of life [3,4,5]. The SMA incidence is approximately 1 in 10,000 live births, while the carrier status incidence varies by ethnicity and ranges from 1 in 54 [2,6].

Due to the dynamic development of diagnosis and the targeted treatment of SMA, the clinical pattern of the disease has changed [7,8]. Previously known classifications into types SMA1, SMA2, and SMA3 have stopped being relevant [1,9]. From a certain date, it has become more common to use division depending on the children’s functional level such as sitters, no-sitters, or walkers [10]. SMA patients undergoing treatment not only regain previously lost functions but also are able to achieve new milestones [3,6,8]. It is becoming increasingly common to define this disease as chronic instead of progressive and untreated. Thus, it becomes necessary to observe long-term changes in the new clinical pattern of SMA [9,11]. Validated functional scales are useful in this regard [12,13,14].

The generally available literature indicates the Hammersmith Functional Motor Scale with expanded version (HFMSE) and Revised Upper Limb Module (RULM) as the most common scales used to assess SMA patients with at least an unsupported sit [12,13,14]. The HFMSE scale is a validated tool to evaluate the gross motor function of SMA patients [13,14]. The RULM scale is used to determine the upper limb function. It is a revised version of the Upper Limb Module including additional tasks to decrease the possibility of a ceiling effect in stronger patients [12,15]. 

Until now, only a few articles investigating the relationship between upper limb function and gross motor function in untreated SMA patients have been published [9,16]. However, there is still a lack of publications including the relationship between structural changes such as cervical rotation, trunk rotation and side trunk shortening, and upper limb function. 

The aim of the study was to examine the upper limb function in patients with spinal muscular atrophy and the relationship between the upper limb function, gross motor function, and structural measurement.

## 2. Materials and Methods

The study had a prospective character and focused on 12 months of observation with children examined at the time of enrollment and followed-up at the end of this period. The data were collected between July 2020 and October 2022.

### 2.1. Materials

Twenty-six patients at the Department of Developmental Neurology, Heliodor Święcicki Clinical Hospital of the Karol Marcinkowski University of Medical Science in Poznan (Poland) were recruited into the study. One child was excluded from the research due to an interruption during the first examination. A total of 25 of the included patients aged 5 to 20 years (Me = 6; Q25–Q75 = 6–12) had a genetically confirmed diagnosis of SMA type 2 or type 3 and achieved at least the unassisted sit function. Fourteen out of this group were females and eleven were males. The patients enrolled in the study were undergoing pharmacological treatment with nusinersen (14 children) and risdiplam (11 children). The study included the division of patients into sitter and walker groups according to the milestone achievement. Detailed characteristics of the participants are presented in Table 1. 

### 2.2. Methods

To assess the relationship between the upper limb and gross motor function and the correlation between upper limb function and structural measurement, the following tools were included.

The Revised Upper Limb Module (RULM) was used to assess the function of the upper limb [12]. The children were examined in a sitting position using a specific test set consisting of coins, a Ziploc container, a button light, and weights of different masses. The test consists of 20 tasks with the entry item (A) serving as functional class identification, which is not included in the total score. Each of the items can be performed three times. Eighteen of them were scored from 0 to 2 points, where 2 points means the task was completed correctly, 1 point means the task was completed with compensation/or less effective, and 0 means the task was not completed. One question with a Ziploc container was scored from 0 to 1, where 0 means a failure to perform and 1 is the correct performance of the task. The total RULM score to obtain is 37 points, and the task evaluates functions such as bringing hands from their lap to the table, completing the path bringing the car to the finish line without stopping or taking the pencil off the paper, picking up coins, placing a coin into cup, reaching to the side and touching the coin, pushing a button light with one hand, tearing paper, opening a Ziploc container, raising a 200 g cup to their mouth, lifting 200 g and 500 g and bringing it from one circle (midline to outer circle), lifting 200 g and bringing to the outer circle on the opposite side, bringing a 500 g sand weight from their lap to eye level, shoulder abduction with and without 500 g and 1 kg of weight, and shoulder flexion with and without 500 g and 1 kg of weight [12,16].

The gross motor function of each of the participants was evaluated with the Hammersmith Functional Motor Scale–Extended (HFMSE) [13,14]. The scale consists of the basic version (20 items) and the supplementary module (13 items). Each question can be scored from 0 to 2 points, where 2 points means the task was completed correctly, 1 point means the task was completed with compensation, and 0 means the task was not completed. The items included functions such as sitting with hands to head, rolling supine to prone, rolling prone to supine, propping on forearms and extended arms, crawling, standing, or walking. The supplementary module consisted of more difficult tasks such as squat, jumping forward, and ascending and descending stairs. The maximum number of points is 66 [13,14]. Both the RULM scale and the HFMSE scale were previously validated [12,17].

The last part of the examination focused on physical impairments (structural parameters). The study included parameters such as the cervical rotation test (CR), supine angle of trunk test (SATR), and side length trunk difference [4,10,18]. CR was measured using a plurimeter (Rippstain Inclinometer) and presented the range of cervical spine rotation. SATR using a scoliometer evaluated the angle of upper (SATR-U) and lower (SATR-L) trunk rotation. The side length trunk difference was tested in a sitting position after previously marking anatomical known surface landmarks of the inferior angle of the scapula and posterior superior iliac spine [19]. The length between these two points on the same side was checked using a tape measure. Detailed characteristics of the functional scales and physical impairments were published in our previous study [10,20]. In order to better present the study protocol, the flowchart presented below was created (Figure 1).

The study was conducted in accordance with the Declaration of Helsinki [21], and approved by the Ethics Committee of Poznan University of Medical Sciences Bioethical Committee (no. 1035/19).

### 2.3. Statistical Analysis

All test results were analyzed using Statistica 12.2 Software by StatSoft (Kraków, Poland). Due to the relatively small sample, the distribution of data for the interval variables differed from normal (Shapiro–Wilk and Lilleforse tests), thus, as for all ordinal variables, the results were shown as the median and quartiles [Me (Q25–Q75)]. Non-parametric statistical tests were used (i.e., U Mann–Whitney test) to assess the differences between the two groups, and Spearman’s rank correlation (rs). For time variability, a sign test was used. In all of the tests, the level of significance was set as *p* < 0.05. 

## 3. Results

The study analyzed the patients’ upper limb, motor function, and structural parameter changes between the initial examination and evaluation after a 12-month period. 

### 3.1. Individual Results in RULM, HFMSE, and Structural Parameters

Two of the examined patients in the sitter group obtained the maximum RULM score at the first examination, thus having no possibility of improvement during the second observation. Additionally, among all of the participants, three out of four patients in the walker group achieved the maximum score in the HFMSE part 1, thereby also having no opportunity to improve this score. None of the patients had a baseline maximum score in the total HFMSE score. A detailed description of the results of the scales and structural parameters is presented in Table 2. Due to the relatively small walker group (*n* = 4), only the min–max value without a median score is given in the table.

The largest range of improvement was visible in the RULM scale. Patients improved by an average of 3 points after a 12-month follow-up. On the HFMSE scale, participants showed progress by an average of 2 points on the first part of the scale and in the total score. However, in the case of the sitter group, the median value dropped in the first part of the scale while remaining at the same level in the total results, which was not observed in the walker group. Additionally, when evaluating the structural parameters, most of them showed no significant changes with the exception of the side length trunk difference. The average value of the difference increased from 2.25 to 3 cm, reaching a maximum difference of up to 13 cm from one child in the sitter group in the second examination.

### 3.2. Comparison of the Changes in RULM and HFMSE in All Participants and Functional Subgroups (Sitter, Walker)

Subsequently, using the RULM scale and the HFMSE scale, it was verified which patients showed improvement, which had deterioration, and which presented unchanged function. To elaborate the results, the changes for all participants as well as with the sitter and walker groups were analyzed. The results are presented in Table 3.

The upper limb function (expressed in the RULM total score) improved more in the whole group than the gross motor function evaluated with the HFMSE. It was also true for the subgroups divided according to the highest function achieved, though exact statistical analysis was not possible for the walker group due to the too small number of children. The improvement was also more expressed in the HFMSE total score than in part 1.

### 3.3. Difference in RULM Items

Another essential point of the study was the insightful analysis of all RULM scale items. It was verified as to in which tasks the highest improvement, deterioration, or stabilization of function occurred. Table 4 presents the number of children in whom such changes after a 12-month period were observed. The evaluation included all participants as well as the sitter and walker groups. 

All participants presented function stabilization in the majority of items after 12 months of observation. However, particular elements showed distinct changes. The highest improvement was observed in the tearing paper task (H). The increase in this function included up to 11 patients. Equally important items were shoulder abduction function with 500 g weight (P), where progression was seen in eight participants. The difference was also observed in the shoulder abduction function with 1 kg weight (Q) and shoulder flexion with 500 g weight (S), but only in the whole group and sitters. The only example of the task with marked deterioration rather than improvement was the shoulder flexion with 1 kg weight (T). This change affected three patients.

### 3.4. Correlation between Measured Parameters

Finally, the last analyzed issue was to check the relationship between the upper limb function, gross motor function, and physical impairments during the first and second examination as well as the relationship between the physical impairments themselves. Due to the relatively small walker group (*n* = 4), the correlation was not analyzed in this group. The results are presented in Table 5.

Both in the first and the second examination, a strong correlation was observed between the RULM total score and the HFMSE total score concerning all participants (first examination Rho = 0.927; second examination Rho = 0.929) and those in the sitter group (first examination Rho = 0.928; second examination Rho = 0.911). Furthermore, considering the RULM total score and structural parameters during the initial examination, it was seen that the total RULM score interacted with right-side cervical rotation (CR-R) and amounted to = 0.408. This change affected all patients and was not observed in the second test. According to the analysis of all patients in the second study, it was noted that the higher the value of the lower trunk rotation (SATR-L), the lower value of the RULM total score (Rho = −0.397). This was not observed for the sitter group. Additionally, it was seen that the shortening of one side (side length trunk difference) had a negative impact on the RULM total score in all participants (Rho = −0.571) and the sitter group (Rho = −0.487).

Subsequently, according to the analysis of the first part of the HFMSE scale, it was observed that the increase in the value of cervical rotation leads to higher improvement in the HFMSE part.1 score. This change is seen in all participants and concerns the right side in both examinations (Rho = 0.404; Rho = 0.415) and only the left side in the initial observation (Rho = 0.398). In addition, during the second examination, a strong negative correlation was noticed between the HFMSE part 1 score and the side length trunk difference in all patients (Rho = −6.676). Moreover, as a result of convergence between the HFMSE total score and the structural parameters, a strong relationship was seen with a side length trunk difference (Rho = −0.99 in all participants, Rho = −0.642 in the sitter group) during the second examination and a medium correlation was observed with CR-R in both examinations (Rho = 0.430; Rho = 0.417), and CR-L only in the initial observation (Rho = 0.416).

Furthermore, in comparison to the structural parameters in the first examination, it was seen that the greater the SATR-U and SATR-L, a greater side shortening was presented in all patients (Rho = 0.632; Rho = 0.590) and the sitter group (Rho = 0.573; Rho = 0.530). In contrast, in the second examination, it was demonstrated that with the decrease in CR-R, the side differences started to increase in all participants (Rho = −4.04). However, it is worth mentioning that the correlation was different for the side length difference and CR parameters during the first examination. The positive correlation was seen in comparison with the CR-R parameter (Rho = 0.276), and a negative relation was observed in the CR-L value (Rho = −0.314).

## 4. Discussion

The purpose of our 12-month follow-up was to assess the upper limb function in SMA patients and to verify its relationship with general motor function and the structural parameters. The examined patients were undergoing pharmacological treatment and came from a single ethnic and cultural area. 

Our results showed that patients demonstrated greater improvement on the RULM scale (18 patients) than the HFMSE scale (13 patients), with an average of 1 point on each scale. The range of improvement was compatible with studies conducted on treated SMA2 and SMA3 patients, which were published by Coratti and Pane [22,23]. In both articles during the 12-month observation, the children improved by 1.59 points (Coratti) and 1.2 points in SMA2 and 0 points in SMA3 (Pane) on the RULM scale. In the case of the HFMSE, the increase was 1.9 points according to Coratti and 1.6 points in SMA2 and 0.9 points in SMA3 in Pane’s publication. However, only a few papers reported an improvement in the treated SMA patients, whereas the data were dominated by the deterioration ranging from 0.13 to as much as 3 points on the RULM scale and about 0.5 points on the HFMSE scale [9,11,24,25,26]. This is most likely due to the later initiation of treatment and the large deficits that have already occurred. More importantly, in the natural course of the disease, Coratti noted a deterioration of function in HFMSE by 0.54 points, while in the RULM it was by only 0.13 points, which may indicate a slower rate of loss function in the upper extremities [9]. Stępień et al. also observed that improvements in the patients’ motor function were associated with improvements in upper trunk muscle strength, which may confirm that improvements in the RULM scale apply to a larger group of patients and are more rapidly noticeable than in the HFMSE [4]. Our study also demonstrated that the HFMSE total score improved more than HFMSE part 1 score. This may be due to the fact that many of these children achieved a relatively high level of gross motor function and thus the improvement was not as probable as the improvement in more complicated tasks. Hence, it is worth mentioning that considering the HFMSE scale, three out of four patients in the walker group achieved the maximum result in the first part of the HFMSE scale, thus having no chance to improve this result after a 12-month observation. A similar situation was visible in the RULM scale, where two patients from the sitter group obtained the maximum baseline score.

On the other hand, these results differed from our previous publication, where an improvement of four points in the first part and a six point total score was shown at the 10-month follow-up [10]. However, that study also included young patients who were both diagnosed and received treatment relatively early, sometimes just after birth, and showed a surprisingly high improvement of nine and 13 points, which agreed with Pane’s assumption that patients show a significantly higher improvement in function in the first 12 months than at 24 [10,23]. Such an early diagnosis and treatment protected those children from both large-scale muscle loss and postural deformities. Nevertheless, for our group of patients, the time from the start of treatment was much longer than in these cases.

Correct function of the upper limbs could not develop until the muscles had adequate strength. Upper limb defects were developmentally smaller due to the higher density of innervation and the loss of smaller motor units without a loss of total function. In older children, entrenched structural abnormalities are very difficult if not impossible to reverse, hence the chance of improved function is less (even with strengthening muscles).

Another essential point of our study was the analysis of individual items of the RULM scale. The results demonstrate that the majority of patients showed stabilization of individual tasks, but some items such as paper tearing and shoulder abduction and flexion improved. Pera’s study on a group of untreated SMA patients showed that patients under the age of 5 years obtained improvements in most RULM parameters, except for tearing paper or shoulder abduction or flexion with weights, which just happened to show the greatest improvement in our patients [24]. This could perhaps be a result of the treatment and the overall strengthening of the muscles performing these activities. In addition, the difference was apparent in the improvement of patients in RULM items testing similar functions. Taking into account the function of shoulder abduction in the case of the “O” item, three patients improved, while in point “P” (shoulder abduction with 500 g weight) the improvement referred to eight patients. A similar relationship applied to the flexion function where two patients improved in the “R” item, and five in the “S” item with shoulder flexion with a 500 g weight. This is due to the fact that most patients maintained the previously achieved, simpler function (in this case “O” or “R”), achieving a new, more complex function “P” or “S”.

Finally, the last part of our study was the analysis of the correlation of upper limb function with functional assessment and structural parameters. The results demonstrated that the RULM had a strong convergence with the HFMSE scale (Rho > 0.900). This is consistent with Coratti’s publication, whose study showed a correlation of 0.730 in SMA2 patients and 0.787 in SMA3 patients [9]. The RULM and the HFMSE score were also related to structural parameters, which, however, is hard to compare with other publications, since to the best of our knowledge, no similar studies have been conducted to date. Our data showed that RULM had a positive correlation with cervical rotation during the first examination, which could be consistent with the Stępień study that showed a significant relationship between neck mobility and upper limb joint mobility [4]. The increase in upper limb contractures limited the range of neck rotation, which may imply a deterioration in hand function and thus global motor function, which could also confirm the positive correlation of CR with the HFMSE. SATR has also shown a negative relationship with the RULM score, which can also be related to the above publication, pointing out that children with greater thoracic deformity show lower muscle strength values. Additionally, we proved in our previous publication that SATR was negatively correlated with the HFMSE score. In this study, we observed older children in whom the structural changes did not progress as quickly as in younger children; also, some of them functioned relatively highly and therefore showed no improvement [4,10]. The structural parameter results also demonstrated that the shortening of one side (side trunk length difference) had a negative relationship with the RULM and the HFMSE total score. Moreover, the greater the shortening of one side, the greater the SATR and the smaller the CR. As can be seen, these parameters are closely related to each other, which is in line with the assumption of Stępień et al. The authors noted the relationship between the magnitude of scoliosis, and the severity of thoracic deformity and pelvic obliquity, which may suggest a progression in the shortening of the length of the side, leading to the deterioration of both upper limb function and overall functional assessment [4]. In our study, we also observed a difference between the correlation of the side length difference and the CR parameter. The Rho value for CR-R and the side length difference was 0.276, while there was a negative correlation of −0.314 for the CR-L parameter. This may be due to the fact that 23 of the 25 patients had the right dominant arm, which may have influenced this result. 

These results show that the presence of structural parameters can impair fine and gross motor function. Therefore, it may represent a significant therapeutic target, which the scale score will not show directly. Therefore, it is important not only to evaluate how the individual scale items are performed, but also whether the lack of improvement in these items is due to worsening the structural parameters. It is also known that the treatment makes positive changes in the patient’s functional level and the recovery of muscle strength. Perhaps a 12-month follow-up is not sufficient time to regain certain motor functions, as they are not directly due to the strength alone, but also to neuromuscular coordination, which also confirms the need to conduct functional physiotherapy. It is worth noticing that all of the SMA patients in Poland are rehabilitated according to the standards based on the International Standards of Care Committee for Spinal Muscular Atrophy created by experts in 2007 and actualized in 2018 [27,28,29]. There is also evidence pointing to the therapeutic importance of exercises promoting functional enhancement [30]. Thus, it is also worth considering the importance of structural parameters in the implications for physiotherapy. It is also necessary to inform other specialists of the results of our research. Because SMA is a disease that requires multiple health care professionals, it is essential to promote interdisciplinary work [31]. 

Statistically significant correlations were sometimes present across the entire investigated group, but not for sitters only. It may be hypothesized that the walkers mainly influenced them, which in turn may be due to more harmonic development (less deficits, especially in the lower part of the body). In sitters, some developmental restrictions mainly present in the lower limbs may cause the lack of correlations of structural parameters with the total score in the RULM and the HFMSE, although the correlation between both scales used was still present. 

The RULM scale does not take into account the child’s age, hence the deficit in a given function is not necessarily due to the disease, and the change may be due to maturation rather than a reversal of lesions. This is in line with Coratti’s research, which noted that the highest rate of decline was observed in patients aged between 10 and 14 [11]. This was also confirmed by Pera, who estimated the largest decrease between the ages of 5 and 14 [24]. Mazzone himself, the author of the scale, also highlighted the need to assess the impact of age, contractures, and gender on individual performance on this scale. The scale well captured the deterioration seen in the natural course of SMA, while it did not best show the change in children who achieved high scores due to treatment from the beginning.

Until now, before the introduction of treatment, there was no possibility of maintaining the children’s functional level. There was no rehabilitation plan, only a plan to maintain the status quo. Currently, we are in a transitional moment when the treatment started to be introduced, but each patient received the drug at a different point in time, and it is difficult to compare children who received the treatment after many years of illness with those who received it on the first day of life through screening tests. It is necessary to continue to observe the effect of treatment on the clinical pattern of the disease in order to adjust the appropriate rehabilitation.

Our study had some limitations. The study group was relatively small and quite heterogeneous. However, other researchers have also struggled with conducting analyses on equally small groups [4,32,33,34]. Nevertheless, it would be necessary to expand the group, especially the walker subgroup, which in our case only had four participants. Due to this, it could be helpful to extend the study to more research centers. Furthermore, future research should also consider assessing the impact of SMA on the psychological and social parameters. Moreover, it would be appropriate to extend the duration of follow-up for additional years, as this will allow us to assess whether the rate of functional recovery will be more noticeable after a longer period of time.

## 5. Conclusions

At the 12-month observation in patients with spinal muscular atrophy, the improvement in the upper limb were more pronounced than the overall improvement in motor function. In addition, the older patients, who were more stable, presented less incremental structural changes. Moreover, persistent structural changes negatively affected both the upper limb function and gross motor skills in this group of patients. 

## Figures and Tables

**Figure 1 biomedicines-11-01005-f001:**
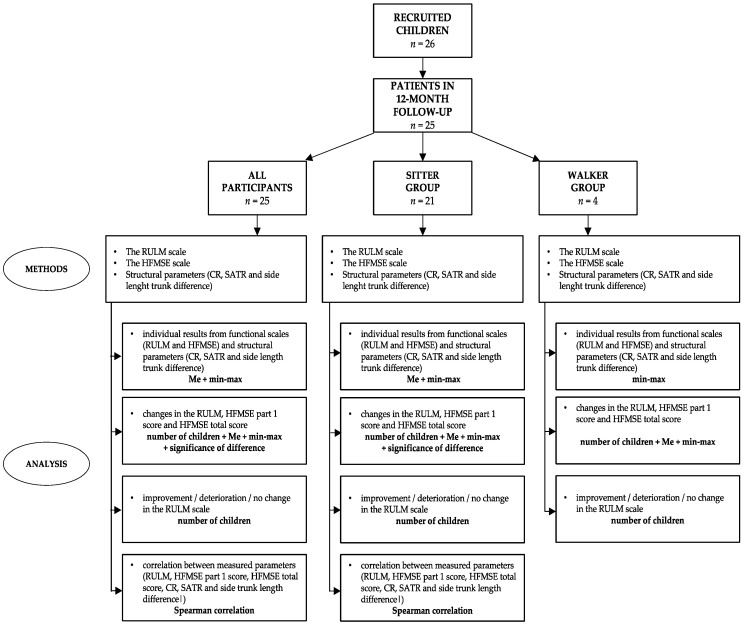
Flowchart with the applied methods and analysis.

**Table 1 biomedicines-11-01005-t001:** Study group characteristics.

PARAMETERS	PARTICIPANTS *n* = 25
Age (years):	
Median (Q25–Q75)	8 (6–12)
Age at the start of treatment (years):	
Median (Q25–Q75)	5 (3–11)
Sex:	
Female	14
Male	11
SMA type:	
SMA2:	18:
Sitter	18
SMA3:	7:
Sitter	3
Walker	4
*SMN2* copy number:	
2 copies	1
3 copies	19
4 copies	5
therapy used:	
Nusinersen	14
Risdiplam	11
child’s functional level	
Sits unassisted	6
Rolls over	6
Crawls	4
Stands with assistance	3
Stands unassisted	2
Walks with assistance	1
Walks unassisted	3
rehabilitation intensity:	
1–3 times a week	4
4–6 times a week	11
Everyday	10
The duration between the start of treatment to the first examination (months):	
Median (Q25–Q75)	23 (10–36)
The duration between the diagnosis to the start of the treatment (months):	
Median (Q25–Q75)	51 (28–95)
Follow-up duration	
Median	12 months

**Table 2 biomedicines-11-01005-t002:** Results from the functional scales and structural parameters during the first and second examination.

PARAMETER	ALL *n* = 25		SITTER *n* = 21		WALKER *n* = 4
	Me	Min–Max	Me	Min–Max	Min–Max
FIRST EXAMINATION					
RULM total score	26	12–37	25	12–37	29–36
HFMSE part 1 score	24	3–40	21	3–37	24–40
HFMSE total score	26	5–63	23	5–45	26–63
CR-R	70	30–85	70	30–85	70–85
CR-L	80	40–90	75	40–90	70–90
SATR-U	5	0–10	5	0–10	0–5
SATR-L	5	0–15	5	0–15	0–5
Side length trunk difference	2.25	1–7	2.25	1–7	1–5
SECOND EXAMINATION					
RULM total score	29	13–37	27	13–37	31–37
HFMSE part 1 score	26	5–40	20	5–45	26–40
HFMSE total score	28	6–66	23	6–46	28–66
CR-R	70	50–90	70	50–90	80–90
CR-L	80	40–90	80	40–90	70–90
SATR-U	5	0–15	5	0–15	0–5
SATR-L	5	0–10	5	0–10	0–10
Side length trunk difference	3	0–13	3	0–13	0–2

Abbreviations: RULM total score—Revised Upper Limb Module total score; HFMSE part 1 score—the result of the first part of Hammersmith Functional Motor Scale–Extended; HFMSE total score—the total result of Hammersmith Functional Motor Scale–Extended; CR-R—right cervical rotation; CR-L—left cervical rotation; SATR-U—upper trunk rotation angle; SATR-L—lower trunk rotation angle; Me—median; min–max—minimal and maximal value.

**Table 3 biomedicines-11-01005-t003:** Changes in the RULM and the HFMSE scales after 12-months of observation in all of the studied children and in the functional subgroups (sitter, walker). Statistically significant difference is marked in bold.

DIFFERENCE IN FUNCTIONAL SCALES BETWEEN EXAMINATIONS	ALL PARTICIPANTS *n* = 25			SITTER *n* = 21			WALKER *n* = 4		
	RULM Total Score	HFMSE Part 1 Score	HFMSE Total Score	RULM Total Score	HFMSE Part 1 Score	HFMSE Total Score	RULM Total Score	HFMSE Part 1 Score	HFMSE Total Score
Improvement	18	11	13	14	10	11	4	1	2
Deterioration	9	9	11	5	9	9	0	0	2
No change	2	5	1	2	2	1	0	3	0
Median Difference	1	0	1	1	0	1	1.5	0	0.5
Min–max	−4–5	−5–5	−6–7	−4–5	−5–5	−6–7	1–2	0–2	−1–3
Significance of the Difference:	**Z = 2.500** ***p* = 0.01**	Z = 0.224 *p* = 0.82	Z = 0.204 *p* = 0.84	Z = 1.835 *p* = 0.07	Z = 0.000 *p* = 1.00	Z = 0.224 *p* = 0.82	-	-	-

Abbreviations: RULM total score—Revised Upper Limb Module total score; HFMSE part 1 score—the result of the first part of Hammersmith Functional Motor Scale–Extended; HFMSE total score—the total result of Hammersmith Functional Motor Scale–Extended; median difference—difference between median in the first and second examination; min–max—minimal and maximal value; statistical significance *p* < 0.05.

**Table 4 biomedicines-11-01005-t004:** Number of children with improvement, deterioration, or without change in individual RULM items (A, B–T). The most improved items are marked in bold.

RULM ITEMS	ALL PARTICIPANTS *n* = 25			SITTER *n* = 21			WALKER *n* = 4		
	IMPROVEMENT	DETERIORATION	NO CHANGE	IMPROVEMENT	DETERIORATION	NO CHANGE	IMPROVEMENT	DETERIORATION	NO CHANGE
A	3	3	19	3	3	15	0	0	4
B	0	1	24	0	1	20	0	0	4
C	2	0	23	1	0	20	1	0	3
D	0	0	25	0	0	21	0	0	4
E	0	2	23	0	2	19	0	0	4
F	1	2	22	1	2	18	0	0	4
G	2	0	23	2	0	19	0	0	4
H	**11**	**2**	**12**	**8**	**2**	**11**	**3**	**0**	**1**
I	1	0	24	0	0	21	1	0	3
J	1	0	24	1	0	21	0	0	4
K	0	0	25	0	0	21	0	0	4
L	1	0	24	1	0	21	0	0	4
M	1	0	24	1	0	20	0	0	4
N	0	0	25	0	0	21	0	0	4
O	3	2	20	3	2	16	0	0	4
P	**8**	**2**	**15**	**7**	**2**	**12**	1	0	3
Q	**5**	**2**	**18**	**5**	**2**	**14**	0	0	4
R	2	2	21	2	2	17	0	0	4
S	**5**	**0**	**20**	**5**	**0**	**16**	0	0	4
T	2	3	20	1	2	18	1	1	2

**Table 5 biomedicines-11-01005-t005:** The relationship (Spearman’s test correlation) between the measured parameters. Statistically significant values (*p* < 0.05) are shown in bold.

PARAMETERS			ALL *n* = 25	SITTER *n* = 21
			Rho Value=	
FIRST EXAMINATION				
RULM total score	and	HFMSE part 1 score	**0.928**	**0.930**
RULM total score	and	HFMSE total score	**0.927**	**0.928**
RULM total score	and	CR−R	**0.408**	0.340
RULM total score	and	CR−L	0.391	0.317
RULM total score	and	SATR−U	−0.174	−0.265
RULM total score	and	SATR−L	−0.349	−0.388
RULM total score	and	Side length trunk difference	−0.152	−0.175
HFMSE part 1 score	and	CR−R	**0.404**	0.307
HFMSE part 1 score	and	CR−L	**0.398**	0.323
HFMSE part 1 score	and	SATR−U	−0.230	−0.294
HFMSE part 1 score	and	SATR−L	−0.320	−0.348
HFMSE part 1 score	and	Side length trunk difference	−0.351	−0.293
HFMSE total score	and	CR−R	**0.430**	0.357
HFMSE total score	and	CR−L	**0.416**	0.312
HFMSE total score	and	SATR−U	−0.261	−0.337
HFMSE total score	and	SATR−L	−0.337	−0.372
HFMSE total score	and	Side length trunk difference	−0.349	−0.291
CR−R	and	Side length trunk difference	0.276	0.184
CR−L	and	Side length trunk difference	−0.314	−0.249
SATR−U	and	Side length trunk difference	**0.632**	**0.573**
SATR−L	and	Side length trunk difference	**0.590**	**0.530**
SECOND EXAMINATION				
RULM total score	and	HFMSE part 1 score	**0.934**	**0.918**
RULM total score	and	HFMSE total score	**0.929**	**0.911**
RULM total score	and	CR−R	0.379	0.150
RULM total score	and	CR−L	0.319	0.271
RULM total score	and	SATR−U	−0.239	−0.211
RULM total score	and	SATR−L	**−0.397**	−0.425
RULM total score	and	Side length trunk difference	**−0.571**	**−0.478**
HFMSE part 1 score	and	CR−R	**0.415**	0.221
HFMSE part 1 score	and	CR−L	0.373	0.290
HFMSE part 1 score	and	SATR−U	−0.318	−0.273
HFMSE part 1 score	and	SATR−L	0.337	−0.280
HFMSE part 1 score	and	Side length trunk difference	**−0.676**	−0.598
HFMSE total score	and	CR−R	**0.417**	0.213
HFMSE total score	and	CR−L	0.392	0.307
HFMSE total score	and	SATR−U	−0.332	−0.280
HFMSE total score	and	SATR−L	−0.370	−0.318
HFMSE total score	and	Side length trunk difference	**−0.699**	**−0.642**
CR−R	and	Side length trunk difference	**−0.404**	−0.243
CR−L	and	Side length trunk difference	−0.359	−0.339
SATR−U	and	Side length trunk difference	0.223	0.265
SATR−L	and	Side length trunk difference	0.329	0.396

Abbreviations: RULM total score—Revised Upper Limb Module total score; HFMSE part 1 score—the result of the first part of Hammersmith Functional Motor Scale–Extended; HFMSE total score—the total result of Hammersmith Functional Motor Scale–Extended; CR-R—right cervical rotation; CR-L—left cervical rotation; SATR-U—upper trunk rotation angle; SATR-L—lower trunk rotation angle.

## Data Availability

Not applicable.

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
