# Peer review of "Examination of Upper Limb Function and the Relationship with Gross Motor Functional and Structural Parameters in Patients with Spinal Muscular Atrophy"

_biomedicines, 2023, doi:10.3390/biomedicines11041005_

Round 1
Reviewer 1 Report
[Summary of the manuscript]
In this manuscript entitled “Examination of upper limb function and the relationship with gross motor functional and structural parameters in patients with spinal muscular atrophy”, the authors reported an analysis of 25 spinal muscular atrophy (SMA) patients undergoing pharmacological treatment (14 patients treated with nusinersen, 11 patients treated with risdiplam). The patients were 21 sitters and 4 walkers. The authors examined the Revised Upper Limb Module (RULM), the Hammersmith Functional Motor Scale-Extended (HFMSE) and the structural measurements of these patients, twice (initial examination and re-examination after 12 months). According to them, these patients demonstrated greater improvement on the RULM scale than on the HFMSE scale. In addition, persistent structural changes negatively affect both upper limb function and gross motor skills.
[General comments to the manuscript]
This is a well-written manuscript on the physical therapy examination of SMA patients. In particular, it is great that the authors suggested that skeletal deformities themselves also contribute to motor function loss of upper limbs and whole body, showing that the trunk rotation angle and side length trunk difference were related to RULM and HFMSE.
Even so, I still have some concerns about this manuscript. My concerns are shown below.
[Major issues of the manuscript]
(1) In Table 4, RULM items O, P and Q (All participants).
The item “O” is “shoulder abduction function without weight”. The item “P” is “shoulder abduction function with 500g weight”. The item “Q” is “shoulder abduction function with 1kg weight”. The patient number with improvement in the item “O” is 3, lower than 8 in the item “P” and 5 in the item “Q”. I am wondering why many patients did not show improvement in the item “O”.
(2) In Table 4, RULM items R, S and T (All participants).
The item “R” is “shoulder flexion function without weight”. The item “S” is “shoulder flexion function with 500g weight”. The item “T” is “shoulder flexion function with 1kg weight”. The patient number with improvement in the item “R” is 2, lower than 5 in the item “S”. I am also wondering why the patient number with improvement in the item (R) was so small, compared to those in other items.
(3) The time interval between the start of treatment and the initial examination.
The effect of the treatment becomes apparent soon after the treatment has been just started. However, as the days go by, the effectiveness of the treatment fades.
To interpret the data of the manuscript, the time interval between the start of treatment and the initial examination is essential. The authors should add such information.
(4) In Table 5, Rho values of CR-R/CR-L and side length trunk difference.
In Table 5, Rho value of CR-R and side length trunk difference was 0.276, while Rho value of CR-L and side length trunk difference was -0.314. These values may not be statistically significant, but one is positive, the other is negative.
I am wondering whether these data may reflect the dominant arm or scoliosis curvature type of the patients. However, the authors did mention neither the dominant arm of the patients nor scoliosis curvature types of the patients. It would be better to discuss the effect of the dominant arm or scoliosis curvature type.
[Minor issues of the manuscript]
(1) The treatment effect of drugs and the rehabilitation effect.
The authors said the improvement of RULM (Table 3), but they also suggested the progression of skeletal deformities (the last paragraph of the Results section, Lines 220-223). Readers, who are taking care of SMA patients, would be slightly confused. Thus, it would be necessary to discuss the treatment effect of drugs and the rehabilitation effect.
When the authors would add more discussion about the treatment effect of drugs and the rehabilitation effect, the manuscript would be much more useful to the readers.
Reviewer 2 Report
Dear Authors
Thanks a lot for the opportunity you have offered me to revise the fascinating manuscript "Examination of upper limb function and the relationship with gross motor functional and structural parameters in patients with spinal muscular atrophy".
As a significant strength, this manuscript examines the upper limb function in patients with spinal muscular atrophy and the relationship between the upper limb function, gross motor function and structural measurement. This proposal is a novelty in the field and adds information to the existing evidence in the literature produced in the field.
As a major weakness, the manuscript sometimes needs more details and clarity concerning methodological steps that would help improve the understanding of the manuscript. Therefore, I have suggested some strategies to improve authors' reporting and increase the quality of their work.
Overall, my peer review is a minor revision.
¶MAJOR ISSUES:
#METHODS
*reporting: being an observational study, the authors should have referred to STROBE (https://doi.org/10.1016/S0140-6736(07)61602-X doi: 10.1097/EDE.0b013e3181577511) as a guide for their reporting. This point should be supplemented. Thus, I suggest authors to organize their reporting of methods and results accordingly.
*Helsinki declaration: Please, add the appropriate reference (doi: 10.1001/jama.2013.281053).
*Ethical code: I suggest authors also add the code for their ethical approval in the main text, "Ethics Committee of Poznan University of Medical Sciences Bioethical Committee (no. 1035/19)". This strategy will enhance the transparency of their research.
*Department of Developmental Neurology: please, specify the hospital and country.
*scales. Do the scales adopt presented values of validity and reliability? I suggest authors mention these values from literature (if they exist).
#RESULTS:
* reporting: it should be in line with STROBE. For example, the figure with the flow chart is missing.
*Please divide the results in different sections in accordance with the participants and the outcomes. It will be more clear.
*statistical value: please, add the statistical values in the main text.
#DISCUSSION:
*limits: the authors report several limits. This is very appropriate. I suggest they also reflect on and include: the need to extend the study to more research centres (e.g., multicentric research). Furthermore, future research should consider assessing the impact of SMA on psychological and social parameters by, for example, investigating parameters such as patients' satisfaction and perceived experience of care (e.g., doi: 10.1080/09638288.2018.1501102) with qualitative studies (e.g., focus groups, interviews).
*Implications for physiotherapy: the authors stated, "the presence of structural parameters can impair fine and gross motor function." This is very relevant. I suggest they add appropriate implications for the physiotherapy management of patients with SMA. For example, evidence suggests the importance of therapeutic exercises promoting function enhancement (doi: 10.3390/jfmk3030040). Please, have a look to this reference and integrate it.
*Implications for other healthcare providers: Since patients with SMA are very complex and managed by multiple healthcare professionals, I suggest including implications to inform also other professionals (e.g. nurses). For example, nurses, who are often overloaded with tasks and subject to Unfinished Nursing Care, how could they manage these patients in their routine while remaining effective and efficient? Please, have a look at these references (doi: 10.1702/3129.31103. doi: )and integrate them into the discussion. This point will improve the paper.
¶MINOR ISSUES:
#ENGLISH:
*need for revision of the manuscript: I suggest the authors have the English revised by a native speaker. In fact, there are several typos and inaccuracies (e.g., points, and grammatical errors).
#REFERENCES:
*date: many references are not recent. I suggest including more up-to-date refs on the topic (e.g. from the last 2-3 years).
#TABLES
*Acronyms. Report in full all the acronyms presented in all tables. Moreover, in table 1, sex should be reported with a capital letter. For example, Me, min-max…
#ABSTRACT
*aims of the study: please, add the aim of the study more clearly
*" pharmacological treatment". Please, report the detail (…).
#KEYWORDS
*I suggest adding other words, such as "physiotherapy" and "rehabilitation".
#NUMBERS:
*when you start a sentence with a number is better to report it as a word. Moreover, in general, the number from zero to ten should be reported as a word. Instead, the number from 11 should be reported as a number in the main text.
#METHODS:
*" spinal muscular atrophy". Please, report it as SMA.
*" structural measurements". It is a tricky term. I suggest renaming them as "physical impairments" In accordance with the ICF - The International Classification of Functioning, Disability and Health.
*" plurimeter". What does it mean? Specify.
Round 2
Reviewer 1 Report
My doubts were cleared by the explanations of the authors. I am also satisfied with the revisions that have been made by the authors.
